# Education Intervention Has the Potential to Improve Short-Term Dietary Pattern among Older Adults with Undernutrition

**DOI:** 10.3390/geriatrics8030056

**Published:** 2023-05-17

**Authors:** Samantha Chandrika Vijewardane, Aindralal Balasuriya, Alexandra M. Johnstone, Phyo Kyaw Myint

**Affiliations:** 1Ageing Clinical & Experimental Research Team, Institute of Applied Health Sciences, University of Aberdeen, Aberdeen AB25 2ZD, UK; 2Department of Public Health and Family Medicine, General Sir John Kotelawala Defense University, Dehiwala-Mount Lavinia 10 390, Sri Lanka; 3The Rowett Institute, University of Aberdeen, Aberdeen AB25 2ZD, UK

**Keywords:** dietary pattern, older adults, nutrition education intervention, community-based

## Abstract

Low-cost educational interventions to improve dietary pattern is a pragmatic solution to prevent undernutrition in low- and middle-income countries. A prospective nutritional education intervention was conducted among older adults aged 60 years or above with undernutrition with 60 people in each intervention and control group. The objective was to develop and evaluate the efficacy of a community-based nutrition education intervention to improve the dietary pattern of older adults with undernutrition in Sri Lanka. The intervention consisted of two modules to improve the diversity, the variety of diet, and the serving sizes of food consumed. The primary outcome was the improvement of the Dietary Diversity Score (DDS) and the secondary outcomes were the Food Variety Score and Dietary Serving Score, which was assessed using the 24 h dietary recall. The mean difference in scores between the two groups was compared using the independent sample *t*-test at baseline, two weeks and three months post-intervention. Baseline characteristics were comparable. After two weeks, only the difference in DDS between the two groups was statistically significant (*p* = 0.002). However, this was not sustained at three months (*p* = 0.08). This study concludes that nutrition education interventions have the potential for short-term improvement in dietary patterns in older adults in a Sri Lanka setting.

## 1. Introduction

While strong evidence is existing on the links between poor nutrition and disease, less research is focused on nutrition strategies to support older adults. A lot of research evidence is available from various parts of Sri Lanka showing a high prevalence of undernutrition among the older population [1,2]. A higher prevalence of undernutrition is observed among the community-dwelling older adults despite most of them living under the care of their children.

Current research evidence shows that nearly 50% of the older population is having an unhealthy dietary pattern. Considering the dietary intake of older adults in Sri Lanka, their main energy source is carbohydrates and almost all older adults consume cereals (predominantly rice). Nearly half of the older adults in Sri Lanka consume fruits, and vegetable consumption is more than 80% [3]. About one-fifth of the people do not consume meat or meat products and only one-fifth consume inadequate serving sizes as recommended [4]. Considering the micronutrients which are essential for metabolic processes among older adults, more than a third of adults were consuming the lower-than-recommended dairy products containing calcium, which is essential to improve bone density and prevent osteoporosis [5]. It has been suggested that inadequate access to nutritious food among older adults may result from low financial status, mobility difficulties, and poor awareness of nutritious food and dietary patterns. Addressing these factors may require different intervention strategies for different geographical and/or cultural settings [6].

Nutritional interventions in older adults, which have been conducted in both the hospital and community setting, include oral supplements, dietary advice, counseling, and home visits [7,8,9,10]. Most often, these interventions are conducted among older adults with malnutrition or at risk of developing malnutrition [11]. Giving oral supplements has been shown to improve the nutritional status of adults, and the sustainability of such programs in low- and middle-income countries is questionable due to the cost and resource implications [11].

Longitudinal studies have shown that health education and health promotion activities extend the number of years of health in older adults [12,13]. Current literature suggests that nutritional education interventions improve the dietary pattern by increasing the knowledge of diet among older adults and are effective to improve the risk of malnutrition [9]. Cost-effective models of intervention have been recommended for developing countries using the available evidence from other countries, based on the worldwide success to improve the dietary pattern using health education and behavior change concepts [14,15]. Health education and counseling about nutrition among older adults have been recommended as strategies to reach the target population due to the high literacy among the population of Sri Lanka [16]. However, very little evidence is available from studies conducted in Sri Lanka.

Against this background, the objective of the study was to develop a community-based nutrition education intervention to improve the dietary pattern among older adults and evaluate its efficacy in the short term, (up to two weeks post-intervention) and at three months post-intervention, to investigate whether it is sustainable in a Sri Lanka setting.

## 2. Materials and Methods

A prospective nutritional education intervention was designed to evaluate the outcome of the intervention. We designed this study with separate intervention and control groups. The prospective nutritional education intervention was registered with the WHO and the clinical trial registry in Sri Lanka Medical Association. Ethical approval was obtained from the Ethical Review Committee, Faculty of Medicine, University of Kelaniya (P 123/6/2018). 

The population literacy in both areas was above 90% [16] and these two areas have a similar variation in socio-economic level, health care delivery mechanism and access to almost similar information sources, such as awareness programs conducted by the Medical Officer of Health, Public Health Midwives and Non-Governmental Organizations.

### 2.1. Study Population

The community-dwelling older adults diagnosed with undernutrition in Moratuwa and Kesbewa DS were identified as study participants. They were drawn from a previous study conducted by Vijewardane and others [4]. This baseline study was conducted among 800 older adults in seven Divisional Secretariat (DS) divisions in the Colombo District using a multi-stage cluster sampling technique probability proportionate to the size. 

A composite criterion has been used to define an older person with undernutrition, as described in Appendix A. The criterion included body composition measurements (body skeletal muscle mass and fat mass) and anthropometric measurements (BMI and mid-upper arm circumference) [4]. Two pre-intern doctors and the Principal Investigator (PI) did this assessment to identify older adults with undernutrition in the large baseline survey conducted three months prior to this intervention study.

In brief, the eligibility criteria for the selection of participants were as follows;

i.being an older person >60 years with a diagnosis of undernutritionii.not suffering from cancer or chronic renal failureiii.able to provide informed consent (i.e., no cognitive impairment)iv.being able to communicate in Sinhala, as the proposed intervention was only conducted in Sinhala, as communication itself can affect the outcome of the interventionv.permanent resident in the selected areas, hence facilitating the subsequent follow-up and minimizing loss to follow-up.vi.currently not a participant in a dietary or lifestyle intervention.

Eligible subjects were provided with study information both verbally and in written format and provided informed consent. Participants of the intervention group were given an incentive of 500 LKR as the travel cost to visit the community clinic center, where the intervention was delivered. Participants of the control group were assessed at their place of residence, and they were not provided with the incentive.

### 2.2. Sample Size Calculation 

The sample size calculation was based on the mean value of the primary outcome (DDS score) of 4.4 (SD 0.9), reported in a previous study conducted among older adults in Sri Lanka [17]. In the absence of data from published literature on post-intervention differences in improving dietary scores between the intervention and control group, the projected difference of 0.7 was considered. For community-based studies with a cluster sampling technique considering a design effect of 1.9 and 20% non-response rate, the final sample size reached was 60 in each group.

### 2.3. Sampling Technique

Two DS areas were selected conveniently out of the seven DS areas selected for the larger descriptive study, where a multi-stage cluster sampling technique was used to recruit participants [4]. Older adults diagnosed to be undernourished in Moratuwa and Kesbewa DS divisions were selected as the study and control groups, respectively. The two communities were homogenous when considering the possible socio-demographic factors as indicators. Although the two DS divisions were closer, it was convenient for the administration and follow-up of the participants.

### 2.4. Development of the Nutrition Education Intervention 

To develop the nutrition education intervention, a few preliminary activities were conducted. These were (1) identifying objectives for the intervention, (2) identifying suitable methods for administering the intervention, and (3) the development of activities to fulfill these objectives. These activities were carried out through five steps, detailed below. In step one, a literature review and expert opinion in the field of nutrition, geriatrics and public health were sought to identify the evidence-based nutrition facts to be applied as part of the intervention. In addition, discussions with the experts in these fields helped to identify the intervention design suitable for the available resources, such as funding and timeframe. The intervention was designed according to the Health Belief Model (HBM) principles. It is a recommended evidence-based model for improving older adults’ dietary behaviors and is widely used in nutrition education literature [18,19]. 

In step two, an intervention was designed according to the Food-based Dietary Guidelines published by the Nutrition Division, Ministry of Health, Sri Lanka [20]. The objectives and the content of the intervention package were based on the results of the descriptive study conducted among older adults [4]. The content included nutritional requirements of older adults, nutritional information about food items, myths and misconceptions related to diet and dietary patterns, methods of preparing nutritious meals at low cost according to their financial status, and home gardening as a method for nutritional security. The intervention package delivery channels included interactive discussions, group work and counseling sessions. 

In step three, the intervention was drafted as two training modules. Each module was developed to cover a set of learning objectives. Module one was developed to improve the diversity of the diet and to improve variety of the diet. Module two was developed to improve the serving sizes of food consumed by older adults. 

In step four, for each training module, lessons were planned to cover each objective. Then, teaching/discussion methods suitable for each step were defined. In planning the lessons, several activities were used to cover one objective. Special emphasis was given to selecting activities related to day-to-day food and nutrition which was familiar to the majority of the older adults in the community. Discussion points were highlighted under each activity. Teaching/learning methods were defined for each activity, ensuring participatory approaches, such as group discussions and group activities. 

In step five, a team of facilitators was selected to deliver the intervention package. In selecting them, the following points were considered. Competency in the subject of nutrition, the experience and expertise of the person in communicating nutritional information to the general public and being conversant in Sinhala. The team of facilitators consisted of a Medical Officer in Nutrition, Public Health Nursing Tutor, Public Health Nursing Sister and a Public Health Midwife from the nearby Medical Office of Health area. The Medical Officer in Nutrition, serving the nearby hospital, was invited to participate as a facilitator in the intervention. The others are field officers serving in the Medical Officer of Health area of Moratuwa. One from each category was invited to participate as the facilitator for the intervention. PI has provided all the instructions and required training to the facilitators prior to the intervention sessions.

These training modules were then discussed with experts in the field of nutrition, health education, public health and geriatrics. The finalized module was evaluated by using a validation tool (Appendix A). PI provided relevant training to the team, which covered the content of the modules developed, the learning objectives of each module, activities to be performed under each module, subject matter on nutrition related to dietary patterns to be delivered through each activity, and the teaching/learning method for each activity (Table 1, Table 2 and Table 3).

The intervention was delivered in two sessions over two weekends due to the limitation of resources. All the group activities (Appendix A) were conducted after rearranging the group into sub-groups of 10 persons. Each sub-group was allocated the same period of time to finish group activities, as per the time allocation in Figure 1. At the end of the first module, feedback from the participants was taken to assess their satisfaction with the session and to improve the clarity of the next module (Appendix A). 

Summary of the intervention.

### 2.5. Implementation of the Prospective Nutritional Education Intervention

This study was implemented in three phases. Phase 1 was the pre-intervention assessment; Phase 2 was the Implementation of the intervention and Phase 3 was the post-intervention assessment using the 24-h dietary recall method. The overall aim of the intervention was to improve the dietary pattern among older adults with undernutrition. The schematic presentation of the prospective nutritional education intervention is shown in Figure 1.

### 2.6. Data Collection Instruments to Assess the Dietary Patterns 

An interviewer-administered the 24 h diet recall questionnaire and the multiple pass recall method (Appendix A), which were used to assess dietary intake. The dietary recall was administered by trained data collectors. The diet consumed on the previous day was assessed. If a participant had consumed a special diet on that day (i.e., party, special occasion), diet consumed one day prior to this was assessed. At every possible time, the procedure was carried out in the presence of caregivers to improve accuracy. The participants were shown visual aids (photographs of food servings/food atlas) to assist and improve the accurate reporting of dietary intake. The dietary recall was then analyzed to calculate nutrient scores.

#### Description of Assessment of the Dietary Scores

Three dietary scores used to assess the dietary pattern were (1) Dietary Diversity Score (DDS), (2) Food Variety Score (FVS) and (3) Dietary Serving Score (DSS). DDS measures the count of food groups consumed by summing the number of unique food groups out of the six subgroups namely, 1. cereal, roots, or equivalents (starchy food), 2. Vegetables, 3. Fruits, 4. meat, fish, eggs or alteration, 5. legumes/lentils, 6. milk/dairy products as described in Appendix A, over the specific period of time of dietary recall [17]. The range of scores can typically be from 1–6. DDS was developed by FAO [21] and it was described as a cost-effective tool in identifying older adults at risk of malnutrition by Oldewage–Theron and Kruger in 2008 [22].

FVS measures a simple count of food items consumed over the last 24 h [17]. The range for this score can typically be from 1–15. It was initially used by Krebs Smith and others in 1987 [23].

DSS measures the portions of different food groups (Appendix A) consumed according to the Food-based dietary guidelines of the Ministry of Health in Sri Lanka [20]. The recommended daily dietary servings (individual) are mentioned in Appendix A. According to this calculation, a score of four was assigned for a participant who is consuming the recommended four daily servings of starchy foods, and the scores are calculated for all other food groups, and they were expressed out of a maximum score of twenty. All three indicators were modified over time and proven to be valid in assessing nutritional adequacy in all age groups. Rathnayaka and others have validated and proven them to be effective in using among older populations in lower-middle-income countries, such as Sri Lanka [17].

### 2.7. Data Collection

Data were collected by two pre-intern medical officers and the PI. The baseline survey was carried out in participants’ own homes to collect pre-intervention data on socio-demographics, economic status, nutritional status and dietary pattern. Subsequently, they were invited to participate and attend a community clinic within close reach of their homes to deliver the intervention. During the implementation of the intervention, at the end of each session, verbal/written feedback was obtained from all participants. The process of evaluation of the intervention was conducted by applying the process indicators, such as the number of older adults who participated, and the number of activities conducted as planned. The post-intervention assessment was conducted after two weeks to assess short-term efficacy and again after three months to assess sustainability in their own homes. Participants were provided with financial incentives (500 LKR) to cover transport costs and closely followed up via direct contact by phone calls to avoid a loss to follow-up. 

### 2.8. Data Analysis 

Data were analyzed using SPSS (version 22.0). The unit of analysis was at the individual level. The normal distribution of the variables was checked visually by histograms. The intervention and control groups were compared with regard to demographic and socioeconomic variables, using the chi-squared test. The probability level of 0.05 was taken as the significant level. The mean values of DDS, FVS and DSS, between the intervention and control group, were compared, at baseline, two weeks, and three months after the implementation of the intervention, using an independent sample *t*-test. 

## 3. Results

A total of 120 older adults participated in the study, 60 each in the intervention and control groups. Table 4 highlights that there was no significant difference with regard to age, sex, level of education and monthly income of the older participants between the intervention and control groups at the baseline (*p* < 0.05). The intervention and control groups were comparable in relation to the DDS, FVS and DSS at baseline (Table 5). However, the two groups were not comparable with regard to marital status (*p* = 0.02).

In the assessment of efficacy, 58 older adults (out of 60) in the intervention group and 55 (out of 60) in the control group participated. Table 6 shows that there was a statistically significant difference between the two means of the DDS (primary outcome) in the intervention and control groups (*p* < 0.05) and no statistically significant difference in the secondary outcomes (FVS and DSS) between the two groups at two weeks.

In the assessment of the sustainability of the intervention, 56 older adults in the intervention group and 52 in the control group participated. Table 7 shows there were no statistically significant differences between means of the DDS, FVS and DSS in the intervention and control groups (*p* > 0.05) after three months. 

Table 8 highlights the results of the process evaluation of the intervention, which confirmed there were no significant differences in participation rates between the intervention and control groups (*p* > 0.05) for both sessions. All the participants rated both sessions of the intervention as satisfactory.

## 4. Discussion

There was no significant difference in the mean data between the intervention and control groups, in relation to the DDS, FVS and DSS at baseline. However, there was a statistically significant difference observed in the mean DDS (primary outcome) in the intervention group, compared to the control group at two weeks, but not in the secondary outcomes (FVS and DSS). No sustainable improvement of dietary scores was observed over time (12 weeks follow-up).

The study showed an excellent response rate of 96.7% (58/60) in the intervention group and 91.7% (55/60) in the control groups, during the two-week follow-up. The response rate after three months was 93.3% (56/60) in the intervention group and 86.7% (52/60) in the control groups, respectively. This supports the notion that attrition rates were not a primary driver of these results. 

Indeed, the process evaluation of the intervention indicated that the coverage (participation for the first and second sessions) was adequate as the number of sessions delivered was 100%, as pre-planned. The analysis of the feedback of the participants at the end of each intervention module showed that the participants rated the overall delivery of the intervention as 100%.

As the dietary scores were compared at the post-intervention assessment, the difference in DDS, FVS and DSS were not significantly different between the study and control groups in the pre-intervention assessment. Although socioeconomic and demographic characteristics (except marital status) were comparable between the two groups at baseline, all three dietary scores were higher and even less significant in the control group, which indicated an improved dietary pattern in comparison to the intervention group (Figure 2). Since the control group had ‘better’ scores already at outset, they may also improve their habits by trial participation, and this may cause a reduction of intervention for the intervention group. 

The results indicated a statistically significant improvement in the dietary diversity score in the intervention group when compared to the control group, calculated from the 24 h recall. The participants in the intervention group added more food groups to their diet after the intervention, compared to the control group. This suggests that there was an improvement in dietary pattern in the intervention group, at least in the short term. Evidence is available from all parts of the world about the wide usage of the 24 h dietary recall method to assess the improvement of dietary quality and/or dietary pattern among older adults in the community setting [24,25,26,27]. 

Considering the variety of food items consumed and the serving sizes of food items in each food group, there was no statistically significant improvement in consumption in the intervention group, compared to the control group. However, it was indicated that the nutrition education interventions are effective in improving the dietary diversity score among older adults in the Sri Lanka Colombo district. Although it was statistically insignificant, a trend towards improvement in the variety of diets and serving sizes of food groups were observed in the intervention group compared to the control group at two weeks. 

At the assessment of sustainability, no statistically significant difference was observed between the intervention and control groups, regarding the diversity score of the diet, food variety and dietary serving sizes, despite weekly reminders for the intervention group. Higher food variety and serving size scores were reported in the control group compared to the intervention group. The plausible reason may be due to a better baseline value of dietary scores among participants in the control group. 

A study conducted by Rathnayake and others in Sri Lanka revealed that the mean food variety score, dietary diversity score and dietary serving score of the older population were 8.7 (SD = 1.5), 7.3 (1.2) and 10.9 (2), respectively [17]. In contrast to these findings, in the present study, in the intervention group, these scores were 6.63 (1.70), 3.91 (0.87), and 8.99 (1.86), and in the control group, the scores were 7.23 (2.21), 4.16 (0.99), and 9.61 (2.86), respectively. Higher scores were reported in the former study as it was conducted among institutionalized older adults, and they are offered preset menus according to the nutritional recommendation and are under supervision. Moreover, it was conducted in a broader geographical location (six provinces and twelve care homes) and may widely represent their dietary habits. The present study was conducted among the community-dwelling older adults in Colombo District.

There are many factors affecting the sustainability of these programs from the higher policy level up to the lower implementation level [28]. A good infrastructure with coordination of all these factors is necessary for sustainable intervention programs conducted in the community setting. The dietary pattern of an individual is influenced by their food choices, and changing the dietary pattern of an individual requires a complex multicomponent intervention; therefore, it may not be achievable with the nutrition education component alone. The inclusion of behavior change intervention strategies may increase efficacy [29]. Such an intervention may require a longer period of intervention delivery to produce a considerable improvement in the dietary pattern, and these factors should be considered for the long-term benefit of intervention that targets the change in dietary pattern. Interventions incorporated with behavior change have shown to be more sustainable compared to the interventions that did not incorporate this component. Even with the behavior change component, the socioeconomic status of participants may affect the long-term sustainability of the intervention [28]. Dietary interventions conducted among participants from low socio-economic backgrounds tend to be less sustainable. Our study participants were undernourished older adults and extra effort may be needed to improve their dietary pattern in the long term. Relevant stakeholders must take the responsibility to set the infrastructure, and thereby improve the dietary behaviors of these vulnerable categories of people. According to a community-based study conducted in Sri Lanka, hypertension and musculoskeletal disabilities were significantly associated with undernutrition among women [4]. It is necessary to improve health infrastructure to address these common issues associated with older age while planning interventions to promote improvement in dietary patterns.

Comparison difficulties exist between this study and the other nutrition intervention studies because of the difference in the type of intervention and tools used to assess the outcome. In our study, the primary outcome was the improvement of dietary pattern, and it was assessed using three dietary scores. The study tool applied was a 24 h dietary recall questionnaire. Other studies have used food frequency questionnaires and health questionnaires to assess dietary intake and observation of behaviors. Interventions conducted for a long period tend to be more effective in changing dietary pattern [28]. 

Considering the culture of Asian countries, most of the community-dwelling older adults live with their children and they depend on their children/relatives for food [4]. Food choices and decisions regarding diet are therefore affected by the extended family members who live with these older adults. The habitual preference in the dietary pattern of each individual may not be reflected in our diet assessment. Our study was not affected by economic status, as both groups had comparable incomes. Nearly half of the participants in the study and control groups had no definite income. Poverty is a major influence on food choice, and it would be relevant to explore the impact of diet inequalities on health in future work. 

### 4.1. Strengths

Serving sizes of different food groups were assessed according to the food-based dietary guidelines in Sri Lanka, which were developed for the ageing adult population. The measurements used to assess the serving sizes were familiar to older adults, such as the size of a coconut spoonful, and the size of a box of matches. Visual aids were used in assessing the serving sizes of food items to minimize recall bias. 

In the process of the development of the intervention, several inputs from experts in the field of nutrition and geriatric care were received. The descriptive study conducted to explore the factors associated with undernutrition and dietary pattern provided the sampling framework for purposeful sampling [4]. We used the DDS as the primary outcome and FVS and DSS as secondary outcomes, which were validated to use in the Sri Lankan setting. The intervention was designed to culture, appropriately, using the current evidence base. The quasi-experimental design of the study enabled to deliver the intervention feasibly for the older participants. The well-established public health infrastructure enabled the delivery of the intervention effectively. The acceptability of the intervention was also assessed in the present study, which was necessary to assess the acceptability of any intervention which was proposed to be conducted within the healthcare system. 

### 4.2. Limitations

The 24 h dietary recall method was used in assessing dietary intake and this method may not represent the usual dietary intake of an individual. There is a possibility of recall bias, although the interviews with the older adults were accompanied by a caregiver most of the time. However, this method subjects to less recall bias, especially when using the multiple pass method compared with the other dietary assessment techniques, such as the food frequency questionnaire, and seven-day food diary. We did not explore the dietary scores extensively up to the food group level or at the individual level. If so, we can analyze the improvement in the dietary pattern after the intervention by each food group. Considering the three dietary scores, the dietary pattern of the control group was better than the intervention group at the baseline, but only within a short time frame. Our intervention was conducted for two consecutive weekends and according to a systematic review, interventions that last for five months or more tend to be more effective and sustainable [28]. Due to the limitation of the resources, we were unable to conduct our intervention to that extent.

### 4.3. The Future Direction of Research

Poor dietary pattern is a multifaceted problem. Future research is needed to identify other potential associated factors to improve the dietary pattern, thereby undernutrition among older adults in order to fulfill the service demand with the rapidly increasing population of old age. More cost-effective intervention studies based on behavior change models should have experimented to improve the dietary pattern among the older population. Factors affecting the sustainability of health education interventions should also be explored in a low- and middle-income country setting in order to promote the healthy dietary pattern of older adults. More intervention studies should be conducted with the participation of caregivers as their dietary behavior can influence older adults.

## 5. Conclusions

We have identified the potential short-term efficacy of nutrition education to improve the dietary pattern among older adults with undernutrition in a lower middle-income country setting. The present study triggers evidence to explore more efficacious, as well as sustainable interventions, in the future. Other robust intervention techniques, such as behavioral change models, should also be implemented along with educational programs for a longer duration to make an intervention more efficacious and sustainable. 

## Figures and Tables

**Figure 1 geriatrics-08-00056-f001:**
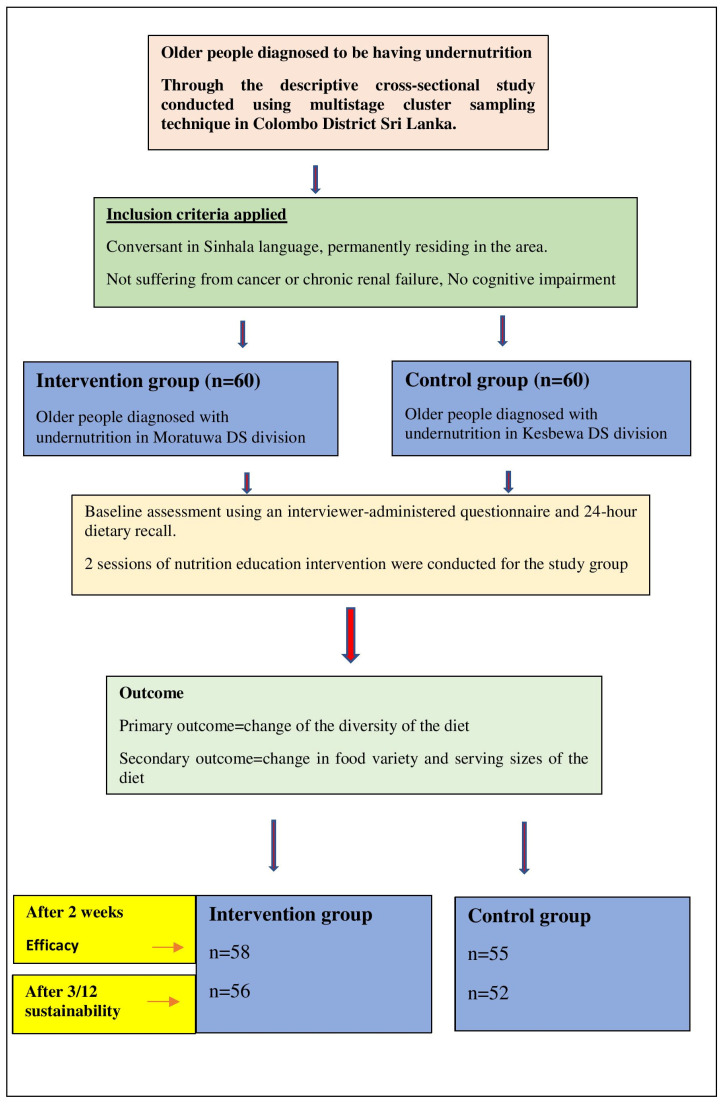
Schematic presentation of the prospective nutritional education intervention.

**Figure 2 geriatrics-08-00056-f002:**
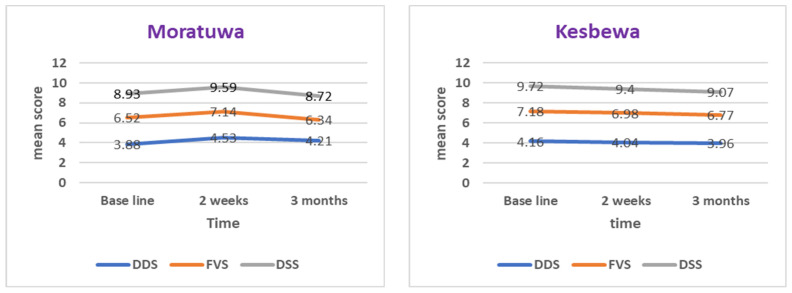
Dietary indicators in the intervention and control groups at baseline, two weeks and three months post-intervention.

**Table 1 geriatrics-08-00056-t001:** Module one-Objective 1: To improve the diversity of the diet consumed by older adults.

Main Discussion Points	Lesson/Activity	Teaching/Learning Method	Facilitator	Time Allocation
Awareness of the importance of nutrition among older adults.	Importance of nutrition among older adults	Interactive Discussion	Medical Officer in Nutrition	20 min
Identifying six main food groups. Developing the ability to identify food items in each food group.	Group activity 1	Interactive Discussion	Public Health Nursing Sister	20 min
Aware of the importance of food groups in a healthy plate	Functions of food groups	Interactive Discussion	Public Health Nursing Tutor	20 min
Developing the ability to identify low cost, highly nutritious food items according to the individual economic status.	Group activity 2	Interactive Discussion	Public Health Nursing Tutor	20 min
Identifying myths and misconceptions related to the diet. Identifying the effect of myths and misconceptions on nutrition.	Group activity 3	Interactive Discussion and group counseling session	Public Health Nursing Tutor	20 min

**Table 2 geriatrics-08-00056-t002:** Module one-Objective 2: To improve the variety of the food items consumed by older adults.

Main Discussion Points	Lesson/Activity	Teaching/Learning Method	Facilitator	Time Allocation
Identifying physiological change with ageing as a barrier to the dietary pattern. Identifying ways of overcoming the problems related to the physiological change with ageing.	Physiological changes with ageing and nutrition	Interactive Discussion	MO in Nutrition	20 min
Developing awareness about the benefits of having a home garden. Identifying ways of home gardening within the limited space.	Group activity 4 Home gardening as a way to improve food variety.	Interactive Discussion	Public Health Nursing Sister	20 min

**Table 3 geriatrics-08-00056-t003:** Module 2-Objective 3: To improve the serving sizes of the food groups consumed by older adults.

Main Discussion Points	Lesson/Activity	Teaching/Learning Method	Facilitator	Time Allocation
Developing awareness about the concept of a healthy plate.	Group activity 5 Concept of a healthy plate.	Interactive Discussion	MO in Nutrition	20 min
Aware of the diet in Diabetes Mellitus and Hypertension.	Nutrition in chronic disease conditions	Interactive Discussion	MO in Nutrition	20 min

**Table 4 geriatrics-08-00056-t004:** Comparison of selected demographic and socioeconomic characteristics and the dietary scores of the intervention and control groups at the baseline.

Characteristic	Intervention Group	Control Group	Chi-Square Value	Significance
*n* = 120	*n* = 60 *n* (%)	*n* = 60 *n* (%)	(df = 1)	
Age 60–70 years	36 (60.0)	28 (46.7)	2.14	*p* = 0.14
Above 70 years	24 (40.0)	32 (53.3)		
Female	51 (85.0)	50 (83.3)	0.06	*p* = 0.08
Male	09 (15.0)	10 (16.7)		
Married	46 (76.7)	34 (56.7)	5.40	*p* = 0.02
Unmarried/divorced/widowed	14 (23.3)	26 (43.7)		
School education Above grade 6	50 (83.3)	55 (91.7)	1.91	*p* = 0.17
Below or equal to grade 6	10 (16.7)	05 (08.3)		
Retired or never employed	51 (85.0)	52 (86.7)	3.15	*p* = 0.08
Currently employed	09 (15.0)	08 (13.3)		
No income	40 (66.7)	30 (50.0)	3.42	*p* = 0.06
With an income	20 (33.3)	30 (50)		

**Table 5 geriatrics-08-00056-t005:** Comparison of dietary scores of the intervention and control groups at the baseline.

Dietary Score	Intervention Group *n* = 60 Mean (SD)	Control Group *n* = 60 Mean (SD)	Mean Difference (95% CI)	Significance (df = 118)
DDS	3.91 (0.87)	4.16 (0.99)	−0.25 (−0.58 0.08)	t = −1.46; *p* = 0.14
FVS	6.63 (1.70)	7.23 (2.21)	−0.60 (−1.30 0.11)	t = −1.66; *p* = 0.09
DSS	8.99 (1.86)	9.61 (2.86)	−0.62 (−1.4 0.24)	t = −1.42; *p* = 0.16

DDS-count of food groups consumed out of the six subgroups; FVS-simple count of food items consumed; DSS-portions of different food groups consumed according to the food-based dietary guidelines, assessed by 24 h dietary recall.

**Table 6 geriatrics-08-00056-t006:** Comparison of dietary scores of the intervention and control groups at two weeks.

Dietary Score *n* = 113	Intervention Group *n* = 58 Mean (SD)	Control Group *n* = 55 Mean (SD)	Mean Difference	95% Confidence Interval	Significance Df = 111
DDS	4.53 (0.84)	4.03 (0.79)	0.50	0.19 0.80	t = 3.23; *p* = 0.002
FVS	7.13 (1.39)	6.98 (1.78)	0.16	−0.44 0.75	t = 0.52; *p* = 0.61
DSS	9.59 (2.27)	9.40 (2.32)	0.20	−0.66 1.05	t = 0.45; *p* = 0.65

DDS-count of food groups consumed out of the six subgroups; FVS-simple count of food items consumed; DSS-portions of different food groups consumed according to the food-based dietary guidelines, assessed by 24 h dietary recall.

**Table 7 geriatrics-08-00056-t007:** Comparison of dietary scores of the intervention and control groups at three months.

Dietary Score *n* = 108	Intervention Group *n* = 56 Mean (SD)	Control Group *n* = 52 Mean (SD)	Mean Difference	95% Confidence Interval	Significance df = 106
DDS	4.21 (0.75)	3.96 (0.73)	0.25	−0.03 0.53	t = 1.75; *p* = 0.08
FVS	6.33 (1.21)	6.76 (1.76)	−0.43	−1.00 0.14	t = −1.48; *p* = 0.14
DSS	8.72 (1.87)	9.07 (2.30)	−0.35	−1.10 0.45	t = −0.87; *p* = 0.39

DDS-count of food groups consumed out of the six subgroups; FVS-simple count of food items consumed; DSS-portions of different food groups consumed according to the food-based dietary guidelines, assessed by 24 h dietary recall.

**Table 8 geriatrics-08-00056-t008:** Process indicators of the two intervention sessions.

Indicator	First Session (*n* = 60)	Second Session (*n* = 58)
Number participated	60/60 = 100%	58/60 = 96.7%
Number of activities conducted as planned.	4/4 = 100%	2/2 = 100%
Rating of the intervention by the feedback		
Content included in the intervention ^a^	59/60 = 98.3%	57/58 = 93.1%
Style of presentation ^b^	57/60 = 95%	55/58 = 94.8%
Mode of delivery of the lessons ^c^	55/60 = 91.6%	56/58 = 96.5%
Overall rating of the intervention ^d^	60/60 = 100%	58/58 = 100%

^a^ perceived satisfaction of the participant on the content of the intervention, ^b^ style of presentation clear/not as perceived by the participant, ^c^ perceived satisfaction of the participant on the mode of delivery of the intervention, i.e., interactive discussion, counseling session ^d^ perceived satisfaction of the participant considering all the above aspects of ^a^, ^b^, ^c^.

## Data Availability

The data presented in this study are available on request from the corresponding author. The data are not publicly available due to the privacy of participants and ethical reasons.

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
