# Peer review of "Education Intervention Has the Potential to Improve Short-Term Dietary Pattern among Older Adults with Undernutrition"

_geriatrics, 2023, doi:10.3390/geriatrics8030056_

Round 1

Reviewer 1 Report

In the present work, Vijewardane et al. try to explain that education intervention can improve short-term dietary pattern among older people with undernutrition. This topic is interesting, but there are some questions that should be explained.

1. Editing of English language and style is needed. Pease revise the manuscript throughout.

For example,

Line 12 ‘comï¼›TP-044’; Lines 15-16, ‘60 years with undernutrition, 60 in each intervention and control group.’; Lines 18-19 ‘modules to improve (i) the diversity (ii) the variety of diet and (iii) the serving sizes of food consumed.’; Line 24 ‘4.53vs4.03’,……, which are not suitable.

2. Abstract

Please delete (DSS), (FVS).

3. Introduction

Line 34, ‘was’

Line 54, ‘were’

4. Materials and methods

Education experience and gender may be considered for the selection of the participants.

Line 158, ‘Pre-intervention’

Figure 1, some arrows have crossed the words, please revise them.

Line 204, Data analysis, mention p-values.

5. Results

Table 1. should be revised, which is very complicated.

Figure 2, please revise the colour of words.

6. Discussion

Line 330, ‘Strengths’?

Line 348, ‘Limitations’?

Line 360, ‘The future direction of research’?

These are the subsection of Discussion section

7. References, Format of references is not suitable for this Journal.

Reviewer 2 Report

The manuscript presents an important topic. However, it focuses on a two-day educational intervention only, demonstrating that it is not effective in the long term. The authors should justify this with scientific evidence. For behavior change a more robust intervention is needed that is not just educational. This should be discussed in the discussion.

The title should be changed as it does not adequately mirror the content.Please clarify the inclusion criteria. Older adults with cognitive impairments were included? See criteria iii

Please clarify the sample size calculation. It is not explicit.

In the results the authors state that 800 older adults participated, but they only present n table 1, 60 participants per group which gives 120 participants and not 800. This should be explained.

Figure 2 should be in results. Not in discussion.

The authors did not evaluate effectiveness, but efficacy. Please change this. Effectiveness is different from efficacy.

Change the word elderly which is out of use, according to new world recommendations. Use older adults instead of elderly.

Reviewer 3 Report

This manuscript describes the development and evaluation of a community-based nutrition education intervention to improve the dietary pattern among older people residing in Sri Lanka. The premise for the research is that despite health education and counseling about nutrition being recommended strategies for older adults in Sri Lanka, there is very little evidence of such trials, programs or evaluations.

Overall comments: Please see my ratings form. Additionally, the manuscript is in need of sweeping improvements in grammar, punctuation, and formatting.

Specific comments:

Abstract

Line 18: In the text, the intervention content is referred to as 2 modules; in the abstract, it’s listed as 3.

Lines 25-27: The conclusion is an over-statement. The authors should consider revising to comment on the feasibility and acceptability (if measured) of the intervention and that preliminary findings suggest a potential for short-term improvement in dietary patterns.

Introduction

Because the setting is unique and the authors state a scarcity of dietary intervention research among older adults residing in Sri Lanka, the introduction could be greatly improved by focusing on the dietary context and behaviors among older adults in Sri Lanka. There are several references that can be used to support these revisions (Pubmed had several). Reformatting the introduction to highlight the research setting and population will add interest.

Methods

Line 68: DS should be defined prior to using abbreviation.

Line 70: What does “indicators” refer to here? Clarification is necessary.

Lines 68-72: Should this paragraph be moved to the section on Sampling Technique on page 3?

Line 75: What do the authors mean by “almost similar information sources?”

Can you provide more information?

Line 79: How were participants recruited? Can the authors provide a brief description? I understand that this study was somehow linked to a larger study. More information is needed for how this intervention study is linked to the parent study.

Line 81: How were participants diagnosed with undernutrition? Who was responsible for this assessment and when did it occur?

Lines 84-89: The way this reads is that these are all inclusion criteria. I am assuming this is incorrect and requires revision.

Line 94: Can the authors clarify how the sample size calculations were estimated, especially lines 96-97.

Line 96: Citation is incorrectly formatted.

Lines 102-143: This section should be divided into multiple paragraphs to make it easier to read.

Do the authors want to refer to the tables in 2.4.1 some place here?

Line 120: Education modules are referred to as ‘2’ in this section but ‘3’ in the abstract.

Line 109: Can the authors say more about how the intervention was designed according to the Health Belief Model? Were relevant constructs assessed?

Line 107: How did expert opinions inform the intervention design? Who were the experts? What areas of expertise? Are these the same experts referred to

in Line 137? Please clarify and add details to the process.

2.4.1 Summary of intervention – formatting of tables and text is inconsistent. Module 1, objective 2 table is challenging to read.

Lines 157-167: This section is repetitive from what has already been described. Is this necessary?

Were participants in the control group wait-listed for the intervention? How were they recruited for participation? I see that participants were incentivized. Can that information be reported earlier in the manuscript?

Figure 1 formatting requires revision/improvement. How many participants were lost to follow-up at each stage? That information should be listed here.

Lines 173-174: Clarification needed – this doesn’t read as a complete sentence.

Lines 178-189: These primary and secondary outcome measures require far more detail. For example, is the DDS the tool that was developed by the FAO? Information about validity and reliability should be included here. I see that it was culturally tailored by checking the reference – that information should be described in the methodology. What was the process for the 24-hour recall?

Line 182: Who developed the FVS? What food groups or items does it include? Similar question for the DDS.

Line 184: Who developed the DSS? The authors? What does S7 refer to?

Results

Line 226, Table 1: If this was an intervention designed for older adults, and only 60% of the intervention group and 46.7% of the control group had ages between 60-70 years, what were the other ages of participants. Please clarify.

Were there within group differences? Did intervention participants significantly improve their dietary scores?

Line 212: The authors state that a total of 800 older people participated in this study. Elsewhere they state 60 intervention, 60 control. Please correct or clarify.

Where do the authors discuss the feasibility and acceptability data? It seems to be missing from the results section. Similarly, which Health Belief Model constructs were addressed and assessed in the intervention? Can the authors comment on these items?

Discussion

Can the authors comment on whether the different in DDS scores at 2 weeks post-intervention is clinically meaningful. In other words, what is the tangible difference in intake between the groups at 2 weeks post-intervention? Can the authors translate the findings to illustrate the actual difference between a DDS score of 4.53 and 4.03. At face value, these scores don’t seem clinically significantly different or meaningful, despite being statistically significantly different.

Lines 260-264: Where are these process evaluation or feasibility outcomes displayed? These should be displayed in the results section.

The discussion could be greatly improved by comparing and contrasting these findings to the literature, instead of repeating the results section again in the discussion.

Lines 284-286: Please revise for clarity. Further, some of these findings should be compared and contrasted in more detail.

Lines 291-293: Can the authors elaborate? The 2 week post-intervention scores were slightly better in the intervention group, but that was not the case for the 3 month scores.

Lines 350-351: Can the authors describe what they mean when they suggest 24-hour recalls are subject to recall bias? Compared to other dietary assessment methods, 24-hour recalls are subject to less recall bias, especially when using the multiple pass method.

Conclusion

The conclusions are overstated. They are not supported by the modest findings. I do think this research can make a contribution to the field if the authors focus on the importance of developing, testing and optimizing dietary interventions for older adults in Sri Lanka. Revisions of this manuscript should focus on the design and development, feasibility and acceptability, and preliminary outcome measures.

Round 2

Reviewer 3 Report

The revised version of the manuscript is greatly improved. I have a few suggestions for clarification.

Lines 37-45: Thank you for adding this contextual information. Some clarification is needed. You state that nearly half of older adults consume fruits and vegetable consumption is 80%. What does that mean exactly? 80% of the older adult population in Sri Lanka consumes the adequate servings of vegetables? If that is the case, wouldn't they be meeting national recommendations? On line 45, you mention that fruit and vegetable intake is below national recommendations. 

In the same paragraph, you mention there are micronutrients that are essential for metabolic processes among older adults, and then highlight the insufficient intake of dairy products. Which micronutrients are you referring to? Clarification of why low consumption of dairy foods is problematic for older adults would be helpful. 

Were intervention and control participants asked if they were participating in other dietary or lifestyle interventions that would coincide with the current intervention? Was that included as exclusion criteria?

Line 105: I think you mean to write "incentive" and not "intensive." Did control participants also receive an incentive for participation?

Lines 163-167: These added lines seem to be missing words (For example, The MO in nutrition... Also, if these abbreviations are not used in the remainder of the manuscript, perhaps they don't need to be abbreviated. PI is defined in the next paragraph, but should be defined first in this section. 

Line 188: This program or This study was implemented...

Lines 197-198: The dietary recall was administered.. not "done." The second line in this section requires editing - the meaning is unclear.

Line 204: Did you define DDS yet? 

Lines 204-220: Formatting seems to change in this section. My recommendation is to write this section in the same narrative style as you've written the manuscript, and beginning with a brief introduction.

Lines 215-220: These added lines seem to be added without consideration of where they are relevant. This information should be split up and located in the section describing each measure. 

Lines 244: The tables should be numbered and displayed in the order they appear in the results section. 

Since each subsection of the results contains only 1-2 sentences, perhaps the results section should just be one section, without subheadings, and with more descriptive narrative.

What exactly is table 5 displaying. The variables are unclear. What about the content included in the intervention? What about the style of presentation and so on.

Tables should not be referred to in the Discussion section.

The conclusion paragraph is vague and requires more narrative and specificity. 
